# Highly Conducting Surface-Silverized Aromatic Polysulfonamide (PSA) Fibers with Excellent Performance Prepared by Nano-Electroplating

**DOI:** 10.3390/nano14010115

**Published:** 2024-01-02

**Authors:** Ruicheng Bai, Pei Zhang, Xihai Wang, Hengxin Zhang, Hao Wang, Qinsi Shao

**Affiliations:** 1Research Center for Composite Materials, School of Materials Science and Engineering, Shanghai University, Shanghai 200072, China; rcbai@shu.edu.cn (R.B.); 1774853814@shu.edu.cn (P.Z.); zhx17855983252@163.com (H.Z.); 2Institute for Sustainable Energy, School of Sciences, Shanghai University, Shanghai 200444, China; 1315741940@shu.edu.cn (X.W.); wh021317@shu.edu.cn (H.W.)

**Keywords:** aromatic polysulfonamide fibers, electroless plating, electroplating, electromagnetic interference shielding

## Abstract

In this work, bilayer nanocoatings were designed and constructed on high-performance aromatic polysulfonamide (PSA) fibers for robust electric conduction and electromagnetic interference (EMI) shielding. More specifically, PSA fibers were first endowed with necessary electric conductivity via electroless nickel (Ni) or nickel alloy (Ni-P-B) plating. Afterward, silver electroplating was carried out to further improve the performance of the composite. The morphology, microstructure, environmental stability, mechanical properties, and EMI shielding performance of the proposed cladded fibers were thoroughly investigated to examine the effects of electrodeposition on both amorphous Ni-P-B and crystalline Ni substrates. The acquired results demonstrated that both PSA@Ni@Ag and PSA@Ni-P-B@Ag composite fibers had high environment stability, good tensile strength, low electric resistance, and outstanding EMI shielding efficiency. This indicates that they can have wide application prospects in aviation, aerospace, telecommunications, and military industries. Furthermore, the PSA@Ni-P-B@Ag fiber configuration seemed more reasonable because it exhibited smoother and denser silver surfaces as well as stronger interfacial binding, leading to lower resistance (185 mΩ cm^−1^) and better shielding efficiency (82.48 dB in the X-band).

## 1. Introduction

With the vigorous development of the aerospace and intelligent electronics industries, the wide application of electronic communications and integrated electronic devices has brought about many electromagnetic radiation problems [1,2,3,4]. They not only interfere with the normal operation of nearby electronic equipment but also endanger human health [5,6,7]. As a result, electromagnetic interference (EMI) shielding materials have emerged, playing an increasingly important role [8,9,10]. Metals and alloys are traditional electromagnetic shielding materials due to their good electric conductivity. However, their high density and poor flexibility are disadvantages that restrict their further application [11,12,13]. Nowadays, researchers are committed to developing flexible, thin, and lightweight materials with high EMI shielding effectiveness. The surface metallization of synthetic fibers seems to be a good option and has attracted considerable attention. [1,14,15,16,17]

Aromatic polysulfonamide (PSA) fiber is a kind of high-performance fiber with independent intellectual property rights in China [18,19]. Surface-metalized PSA fiber composites have advantages such as being lightweight and possessing high strength, excellent flame retardancy, outstanding chemical resistance, and high electrical conductivity. Therefore, they can be used as conductive and electromagnetic shielding materials in aviation, microelectronics, aerospace, military, communications, and other sectors [20]. Methods for loading these functional metallic nanolayers include sputter plating [21], vacuum deposition [22], airbrushing [23], electroless plating [24,25,26,27,28], and electroplating [29,30]. Among them, electroless plating is the easiest to industrialize because of its facile process and simple equipment [31]. Moreover, electroless nickel [20] and nickel alloy (Ni-P [32], Ni-W-P [24], Ni-Co-Fe-P [1], etc.) plating accounts for the dominant portion of the electroless plating field. This is due to benefits such as low cost, stable plating solution, and great adherence to the substrates. However, the obtained coating has moderate electric conductivity and environmental stability, which is inadequate for application scenarios with higher requirements.

Electrical conductivity is one of the primary requirements for a material to be used as an EMI shielding material [33,34]. Silver is then widely used to construct EMI shielding composites due to it possessing the highest electric conductivity (6.30 × 10^7^ S m^−1^ at 25 °C) among metals [35,36]. Zhou et al. [37] prepared flexible and highly conductive meta-aramid (PMIA) nonwoven fabrics by combining polydopamine (PDA) modification and electroless silver plating. The fabrics had an electrical conductivity of 0.29 Ω/sq and an excellent EMI shielding efficiency (SE) of 92.6 dB. Zhang et al. [38] prepared poly(lactic acid)/silver (PLA/Ag) nanocomposites by combining coating Ag particles and compression molding. Due to the existence of the silver coating, the electrical conductivity and EMI performance of the nanocomposites were obviously improved. Tan et al. [39] prepared flexible wearable material with an ultra-high electromagnetic shielding efficiency of 71 dB by electroless plating Ag on a cotton fiber surface (AG@CFS). It is widely acknowledged that due to the high redox potential and catalytic self-deposition effect of silver, the solution for electroless Ag plating is unstable. Electroplating can effectively avoid these disadvantages and improve the availability and reliability of the plating solution. Ag electroplating systems are mainly divided into cyanide and cyanide-free categories. Potassium cyanide is the best complexing agent for Ag electroplating. However, its extreme toxicity poses a high risk to human health and the environment [40,41]. A variety of complexing agents have been proposed for cyanide-free solutions, such as thiosulfate [42,43,44], 3-mercapto-1-propanesulfonic acid [45], 5,5-dimethylhydantoin (DMH) [40,46,47,48,49], nicotinic acid [50], uracil [51,52], and ionic liquids [53,54]. Among those agents, DMH is a potential candidate because it can form relatively stable anionic complexes with Ag ions and produce a uniform and bright silver coating [46].

Therefore, the present study innovatively proposed a novel etchant-, tin-, and Pd-free activation strategy for PSA fibers, where they were coated via electroless Ni or Ni-P-B alloy plating. Afterward, Ag electroplating was innovatively carried out using DMH as a complex to form an outer functional coating. Electroless plating was a very simple and efficient method with which to form a base layer with the necessary conductivity and speed. At the same time, the subsequent electroplating could avoid the drawbacks of electro-less Ag plating and improve the performance of the composite fiber. In this study, the cyanide-free DMH silver electroplating was disclosed for the first time in the metallization of PSA fibers, enhancing the efficiency and controllability of plating. The morphology, structure, environmental stability, mechanical properties, and EMI shielding performance of the proposed cladded fibers were thoroughly investigated. This was to examine the detailed effects of electrodeposition on both amorphous Ni-P-B and crystalline Ni substrates via scanning electron microscopy (SEM), transmission electron microscopy (TEM), X-ray diffraction (XRD), and other techniques. In particular, herein, much work has been done on the ultrathin slicing of the PSA@Ni@Ag and PSA@Ni-P-B@Ag composite fibers. After precise ultrathin sectioning, the complete coating ring with a double-layered structure was systematically investigated and revealed for the first time under TEM.

## 2. Materials and Methods

PSA fibers were provided by Shanghai Tanlon Fiber Co., Ltd., Shanghai, China. Each monofilament consisted of 3000 fibers with an average diameter of 15 μm. Analytically pure chemical reagents were provided by Sinopharm Chemical Reagent Co., Ltd. (Shanghai, China), including sodium hypochlorite aqueous solution (NaClO), silver nitrate (AgNO_3_), sodium borohydride (NaBH_4_), trisodium citrate dihydrate (C_6_H_5_Na_3_O_7_·2H_2_O), nickel acetate tetrahydrate (Ni(CH_3_COO)_2_·4H_2_O), sodium hypophosphite monohydrate (NaH_2_PO_2_·H_2_O), sodium dodecyl sulfate (SDS), potassium carbonate (K_2_CO_3_), 2-Butyne-1,4-diol (BYD), potassium hydroxide (KOH), dimethyl borane (DMAB), ammonia (NH_3_·H_2_O), 5,5-Dimethylhydantoin (DMH), triethanolamine (TEA), L-histidine (HIS), polyethylene glycol (PEG), sodium chloride(NaCl), sodium hydroxide(NaOH), hydrochloric acid (HCl), and ethanol (C_2_H_5_OH). Deionized water was used to prepare all aqueous solutions.

The de-oiled PSA fibers with a length of 15 cm were immersed in the NaClO-HCl solution (pH = 3.00) at room temperature for 15 min for chlorination modification. The Pd-free activation process reported in our previous study [19] was then performed to introduce Ag nanoparticles on and near the fiber surface via immersion in the AgNO_3_ solution and successive reduction in the NaBH_4_ solution. The bath composition and operating conditions of the electroless plating of the Ni or Ni alloy were elaborately adjusted, and the optimal formulations are demonstrated in Table 1. The acquired Ni alloy layer was designed to have high phosphorus (P) and boron (B) contents. The surface-nickeled PSA fibers with a weight gain of 35 wt.% were gently washed in deionized water and recorded as PSA@Ni and PSA@Ni-P-B.

Ag electroplating was conducted on electrically conductive PSA@Ni and PSA@Ni-P-B fibers. The plating formulation and conditions are shown in Table 2. The stabilizer E consisted of 0.5 g∙L^−1^ L-histidine, 4 g∙L^−1^ 2-Butyne-1,4-diol, 2 g∙L^−1^ triethanolamine, and 0.1 g∙L^−1^ polyethylene glycol. The solvents were ethanol and deionized water with a volume ratio of 3:1. In the electroplating process, the fiber samples were used as the cathode, the pure silver plate was used as the anode, and the Ag coating was plated at a voltage of −0.45 V using the constant potential method. The resulting cladded fibers were softly washed in deionized water and dried in a vacuum oven at 80 °C for 1 h and named PSA@Ni@Ag and PSA@Ni-P-B@Ag.

The corresponding schematic illustration of the procedure for the fabrication of silverized PSA composite fiber via pretreatment, electroless plating, and electroplating is displayed in Figure 1.

The surface morphology of the samples was characterized by scanning electron microscopy (SEM, JSM-7500, JEOL, Tokyo, Japan). The fiber cross section was investigated using a JEM-2010F high-resolution transmission electron microscopy (TEM, JEOL). Selective region electron diffraction (SAED) was also applied for further study. The fibers were pre-embedded in the epoxy resin and cut with a diamond knife into thin slices approximately 30 nm thick. Wide-angle X-ray diffraction (WXRD, D/Max 2200V/PC, Rigaku, Tokyo, Japan) with Cu-Kα radiation was used to observe the crystalline structure of the sample. During XRD measurement, the 2θ Angle ranged from 10 to 90° and the scanning step size was 0.02°. The mechanical properties of the single fiber were tested on an XQ-1C tensile tester (Shanghai New Fiber Instrument Co., Ltd., Shanghai, China) at room temperature based on the Chinese standard GB/T14344 [55] (similar to ISO 5079 [56]). The test was carried out 50 times and the results were averaged for each sample. A thermogravimetric analyzer (TGA, Q500, TA, New Castle, DE, USA) was used for thermogravimetric analysis at a heating rate of 10 °C/min with a temperature range of 25 to 700 °C. The electrochemical stability of coated fibers was measured in a conventional three-electrode electrochemical cell by using a microcomputer-based electrochemical workstation (CHI760E, Shanghai Chenhua Instrument Co., Ltd., Shanghai, China) at room temperature. The working electrode was the sample, while a platinum wire was used as a counter electrode and a saturated calomel electrode acted as a reference. The electrolyte was a 5 wt.% NaCl solution.

The electric resistance of the cladded PSA fibers was measured at room temperature by a digital multimeter with a Kelvin small resistance measurement instrument (Shenzhen Everbest Machinery Industry Co., Ltd., Shenzhen, China). The test was repeated 30 times and the results were averaged for each sample. The magnetic properties of the samples were studied by a physical property measurement system (PPMS, EverCool-II, Quantum Design). The EMI shielding efficiency of the cladded samples was tested using the vector network analyzer (VNA, Agilent, N5244A, Santa Clara, CA, USA) in the frequency range of 8.2–12.4 GHz according to the ASTM D5568-14 standard [57] (waveguide method). The PSA fabric electroplated using the same metallization procedure was cut into a rectangular shape with a size of 22.9 mm × 10.2 mm for measurement.

## 3. Results

### 3.1. Morphology and Structure of PSA@Ni@Ag and PSA@Ni-P-B@Ag

Figure 2a,b illustrate the surface morphology of PSA@Ni and PSA@Ni-P-B, respectively. Figure 2a′,b′ depict the corresponding surface topography of PSA@Ni@Ag and PSA@Ni-P-B@Ag after the successive Ag electroplating process, respectively. The Ni-P-B plated layer consisted of larger particles and displayed higher surface roughness than the Ni layer. However, the successive Ag electrodeposition filled the unevenness and produced a smoother surface, with smaller Ag discoid particles on the Ni-P-B base layer. The Ag-plated layer on the Ni substrate seemed to have larger particles and displayed a more undulating surface. From Figure 2c, it is clear that the Ni-plated layer was a very typical nanocrystalline structure. In contrast, the Ni-P-B alloy layer was amorphous due to the massive co-deposition of the non-metallic P and B atoms. The strong crystallized Ag structure could be found for both composite fibers after Ag electrodeposition (Figure 2c′), completely covering the base layer. Peaks were found at 2θ = 38.12, 44.28, 64.43, 77.47, and 81.54°, which were indexed as the reflections from the (111), (200), (220), (311), and (222) planes of metallic Ag (JCPDS Card No. 4-783), respectively. The diffraction peaks of the (111), (200), and (220) crystal planes of PSA@Ni@Ag were more intense. However, the (311) and (222) diffraction peaks were stronger for PSA@Ni-P-B@Ag. The results showed that the surface morphology and crystalline grain orientation differed for the same Ag electrodeposition on the two different base layers. This indicates the different influences from the different substrate structures.

Figure 3 shows the TEM image of the ultramicrotomed PSA@Ni@Ag fiber. Figure 3a demonstrates a relatively complete ring, indicating a good binding fastness between the whole coating and the fiber substrate. This can be attributed to the riveting effect of the Ag/AgCl interface layer produced by the activation process. Figure 3b,c are the enlarged images of the regions marked in Figure 3a. In these figures, the PSA fiber, the Ag/AgCl interface layer, the Ni layer, the Ag layer, and the outermost embedding resin can be clearly distinguished. The thickness of the Ni layer was about 100~150 nm, while the thickness of the Ag layer was about 250~300 nm. Figure 3d–f are the enlarged images, exhibiting the detailed structure of Figure 3c. The Ni layer was composed of many small grains, and the outer Ag layer was quite dense from the cross-sectional view. A shallow crack can be seen between the Ni base layer and the outer Ag layer after the stress compression and deformation during the slicing process (Figure 3b–f). Figure 3g,h are SAED patterns of the marked regions in Figure 3c. Although a complete diffraction ring was not formed due to the thinness of the coating, both images display a polycrystalline structure. The existence of the (111), (200), (220), (311), (400), (331), and (422) crystal planes of Ag were identified in the diffraction spots of Figure 3g. In addition, the diffraction spots of the (111), (200), (220), (311), (331), and (420) crystal planes of crystalline Ni also appeared, as marked in Figure 3h.

As shown in Figure 4, the TEM image of the PSA@Ni-P-B@Ag fiber exhibited a relatively complete metal ring. This indicates good adhesion between the silver-seeded fiber substrate and the whole coating and the good inner bonding of the bilayer coating, making the plated fiber more resistant to plastic deformation during the cutting process. Figure 4b is the enlarged image of the circular region marked in Figure 4a, where the substrate PSA layer, the Ag/AgCl interface layer, the Ni-P-B alloy base layer, the Ag layer, and the embedded resin layer can be clearly seen. The thickness of the Ni-P-B base layer was about 200~250 nm, while the thickness of the Ag layer was about 250~300 nm. Figure 4c–e are the further enlargement of each layer in Figure 4b, revealing a uniform and dense Ni-P-B layer and an Ag layer with grains randomly clustered together. No cracks were found between the Ni-P-B base layer and the electroplated Ag layer even after suffering the large stress during the slicing process, demonstrating the firm inner binding of the bicoating (Figure 4f). A typically diffuse amorphous halo of the Ni-P-B layer was detected via SAED and the result is displayed in Figure 4g, which is consistent with the XRD result. The grains of the Ag layer exhibited a nano-polycrystalline diffraction pattern in Figure 4h, where the (111), (200), (220), (311), (222), and (420) crystal planes of face-centered cubic (FCC) Ag can be found.

According to Figure 3 and Figure 4, the crystal orientation of the crystalline Ni substrate had a certain resistance to the deposition and growth of the electroplated Ag layer, and the combination between the bilayers was not firm. Therefore, they could be easily separated during the slicing process. The electroless Ni-P-B alloy substrate (with high P content) might be better than the pure Ni substrate. The amorphous structure of the substrate may be the overwhelming advantage because it imposes no lattice-matching pressure that inevitably exists between the crystalline substrate and the epitaxially grown film.

### 3.2. Analysis of Environmental Stability and Mechanical Properties

The PSA@Ni@Ag and PSA@Ni-P-B@Ag fibers were placed in deionized water under ultrasonic vibration (40 kHz, 100 W) for 300 min to verify the mechanical stability of the coatings. Figure 5a,b are the SEM images of the resulting PSA@Ni@Ag and PSA@Ni-P-B@Ag fibers, respectively, from which it can be found that there were no obvious peeling, delamination, or cracks. After exposure to ultrasonication for 300 min, the surface resistance increased slightly from 190 to 235 mΩ∙cm^−1^ for PSA@Ni@Ag fibers and from 185 to 200 mΩ∙cm^−1^ for PSA@Ni-P-B@Ag fibers. This also revealed the remarkable adhesion between the coating and the fiber substrate, as well as the excellent cohesion of the coating. The excellent combination improved the wear resistance of the metalized fibers during processing and utilization and enhanced their applicability. Figure 5c shows the tensile strength of the fibers. Compared with the pristine PSA fiber, the breaking strength of PSA@Ni@Ag and PSA@Ni-P-B@Ag decreased by 7.86% and 6.94%, respectively. This can be attributed to the damage of the fiber molecular chain by the alkaline plating solution, but the decrease was acceptable. Furthermore, the tensile strength of the fibers was not further decreased by the electroplating process compared with the electroless plating process. Figure 5d displays the thermal stability of PSA@Ni@Ag and PSA@Ni-P-B@Ag in the N_2_ atmosphere. The mass loss of the first stage (20~200 °C) was caused by water volatilization. The mass loss of the second stage (400~600 °C) was attributed to the partial decomposition of the polymer molecular structure. The initial decomposition temperature of the PSA@Ni@Ag fiber was 446.4 °C, and that of the PSA@Ni-P-B@Ag fiber was 450.1 °C. Both temperatures were slightly lower than that of the pristine PSA fiber (464.0 °C) [19] due to the damage to the fiber molecular chain during the plating process. The residual mass of the original polymer fiber was less than that of the metalized fibers at 700 °C, which was attributed to the greater heat stability of the metallic part of the latter.

Figure 6a–c are the SEM images of PSA@Ni@Ag immersed in the 5 wt.% NaCl solution, 10 wt.% HCl solution, and 10 wt.% NaOH solution for 24 h for chemical corrosion tests, respectively. Figure 6d–f are the SEM images of PSA@Ni-P-B@Ag after immersion in the corresponding solutions for 24 h. It can be seen that the electroplated samples had excellent corrosion resistance to the saline, strongly acidic, and strongly alkaline solutions. After impregnation for 24 h, there was no significant change in fiber surface morphology. It can also be seen from Figure 6g that there was little change in the resistance values of the cladded fibers after the corrosion test. Good corrosion resistance can effectively broaden the application range in environments with different pH values. The three-electrode method and linear sweep voltammetry were used to scan from −1.20 to +1.00 V to further investigate the corrosion stability of the cladded fibers. Figure 6h shows the anodic polarization curves of PSA@Ni@Ag and PSA@Ni-P-B@Ag in the 5 wt.% NaCl solution. There was no sudden current change within the measurement range. This indicates that the anodic oxidation did not occur for both cladded fibers, confirming their stability in the saline solution.

The analysis of environmental stability and mechanical properties indicated the outstanding interface bonding, corrosion resistance, thermal stability, and mechanical strength of these two types of cladded PSA fibers. This allows the fibers to be used in long-term service in harsh environments.

### 3.3. Analysis of EMI Shielding Performance

After the electroless process, the weight gain of the coating was kept at approximately 35.00 wt.%, while it was kept at approximately 45.05 wt.% after the subsequent electroplating. They were both relatively low but were important for maintaining the lightness and flexibility of the fiber. The thin but dense bilayer coating completely covered the entire fiber, as can be seen from the surface and cross-sectional morphology discussed in Section 3.1. A low electric resistance value of 185 mΩ∙cm^−1^ was obtained for the PSA@Ni-P-B@Ag fiber, while it was 190 mΩ∙cm^−1^ for the PSA@Ni@Ag fiber. The former was lower due to the smoother Ag coating and tighter binding with the base Ni-P-B layer. In Figure 7a, the bulb glowed brightly when the prepared PSA@Ni-P-B@Ag fiber was connected as a wire in a circuit. The magnetic properties of the metalized fibers were also tested, and the M−H curves are shown in Figure 7b,c. The PSA@Ni@Ag fibers demonstrated a certain ferromagnetism degree owing to the ferromagnetic Ni base layer, while the PSA@Ni-P-B@Ag fibers were weakly paramagnetic.

In addition to the electric and magnetic properties of the electroplated fibers, the EMI shielding efficiency (EMI SE) was measured for the Ag-electroplated PSA fabrics using the same metallization procedures, and the results are shown in Figure 7d–f. In general, when using vector networks to measure EMI SE, the total shielding efficiency (SE_T_) is composed of reflection (SE_R_) and absorption (SE_A_), which can be calculated by measuring the scattering coefficient. The EMI shielding efficiency satisfies the following equations [58,59]:R = |S_11_|^2^(1)
T = |S_12_|^2^(2)
(3)$${\mathrm{SE}}_{R}=10log\frac{1}{1-R}$$
(4)$${\mathrm{SE}}_{A}=-10log\frac{T}{1-R}$$
(5)$${\mathrm{SE}}_{T}=10log|\frac{1}{T}|={\mathrm{SE}}_{R}+{\mathrm{SE}}_{A}$$
where, S_11_ and S_21_ are the scattering parameters of fabrics measured by the vector network analyzer, R is the reflection coefficient, and T is the transmission coefficient.

The EMI shielding efficiencies of both PSA@Ni@Ag and PSA@Ni-P-B@Ag fabrics were above 80 dB in the wave frequency range of 8.2–12.4 GHz (X-band). This means that more than 99.999999% of electromagnetic waves could be blocked, and only 0.000001% of waves were transmitted through the fabric. The efficiency value was relatively stable in the entire X band, and the SE_A_ value was about three times the SE_R_ value. The EMI shielding efficiency of the PSA@Ni-P-B@Ag fabric was slightly higher than that of the PSA@Ni-P-B@Ag fabric, which can be attributed to its better electric conductivity. The Ni crystalline base layer was intentionally designed to provide good ferromagnetism. The better magnetic permittivity of PSA@Ni@Ag should produce more magnetic loss and bring better EMI shielding efficiency. However, its real impacts were minimal according to the test results, while electric conductivity may play a crucial role in reaching a higher efficiency value. Both PSA@Ni@Ag and PSA@Ni-P-B@Ag fibers presented excellent EMI shielding performance with low weight, high flexibility, good environmental stability, and high mechanical properties. They are expected to be applied in coaxial cables and EMI shielding textiles to prevent EMI and signal leakage.

Based on the above results and analysis, a reliable EMI shielding mechanism for a single metallized PSA fiber is illustrated in Figure 8a. When the EM waves reached the surface of the fiber, considering the abundant free electrons in the surface Ag layer, part of the EM wave was prevented from entering the fiber through direct reflection. The rest of the EM waves entered the interior of the outer Ag coating and the inner Ni/Ni-P-B coating and were weakened by the conductive loss. Noticeably, considering the different electric conductivities and magnetic permittivity of the Ag layer, Ni layer, Ni-P-B layer, and the PSA substrate, there must be massive EM interfaces of metal/metal and metal/polymer. As a result, large-scale internal multiple interface reflection could occur and further attenuate EM waves. The major EMI shielding mechanism for the woven fabric is then shown in Figure 8b. Interestingly, the fabric was composed of a number of tightly bonded conductive fibers, which not only formed a conductive network but also produced internal multiple reflections for EM waves within the fiber arrays. The woven macrostructure trapped most EM waves inside the fabric, which extended their transmission path and produced a large amount of additional absorption. The internal multiple reflection, along with the internal multiple interface reflection, had a non-negligible contribution to the EMI shielding effect. [60]

## 4. Conclusions

In this work, a simple and effective method is proposed to prepare PSA@Ni@Ag and PSA@Ni-P-B@Ag composite fibers with low weights, excellent conductivity, high flexibility, good strength, and high durability for robust electric conduction and EMI shielding. The PSA fiber was first coated by an inner Ni or Ni-P-B base layer via electroless plating. Then, an outer Ag layer was deposited subsequently via electroplating. The proposed bilayer coatings were relatively thin (within 500 nm) compared to the diameter of the PSA fiber (15 μm). However, it provided sufficient conductive channels for electron transport and presented an extremely low electric resistance for both PSA@Ni@Ag (190 mΩ∙cm^−1^) and PSA@Ni-P-B@Ag (185 mΩ∙cm^−1^) configurations, beneficial to improving the conductive loss of the incident EM waves. Furthermore, the existence of numerous metal@metal and polymer@metal interfaces, as well as the special woven structure, enhanced internal multiple reflection and absorption of EM radiation. The whole cladded fibers exhibited high EMI shielding efficiency (above 80 dB in the X band) with weaker reflection to the atmosphere (below 20 dB) and stronger absorption (above 60 dB). The amorphous Ni-P-B substrate had less interference with the deposition, nucleation, and grain growth of silver atoms during the electroplating process, which also implied a wider range of appropriate current densities for deposition. The proposed PSA@Ni-P-B@Ag configuration exhibited better coating density, flatness, and smoothness, as well as stronger interfacial bonding, which yielded better electric properties and EMI shielding performance. The ferromagnetism of PSA@Ni-P-B@Ag could produce more magnetic loss and bring better EMI shielding performance. However, according to the waveguide EMI shielding test results, its real positive impacts were minimal, while electric conduction might play the most important role in reaching a higher efficiency value.

## Figures and Tables

**Figure 1 nanomaterials-14-00115-f001:**
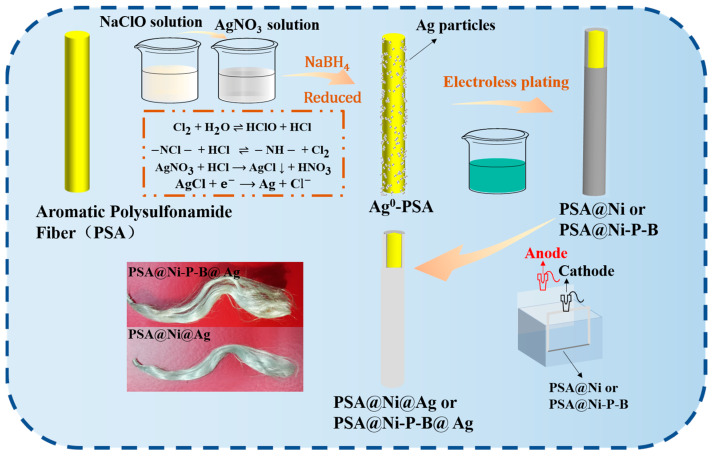
Schematic illustration of the procedure and mechanism for the fabrication of silverized PSA fiber via pretreatment, electroless plating, and electroplating.

**Figure 2 nanomaterials-14-00115-f002:**
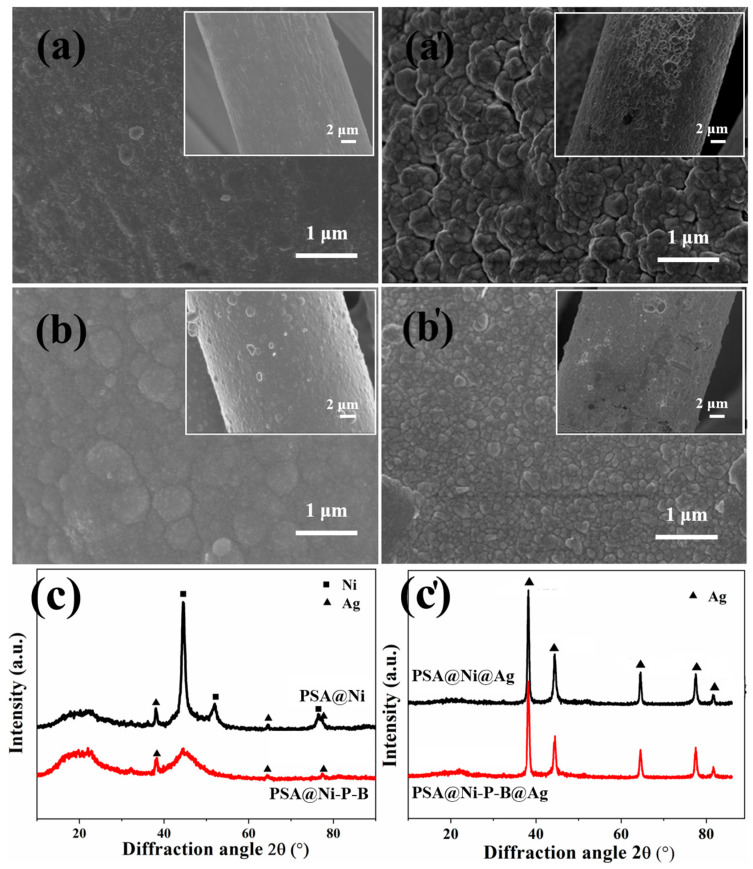
SEM and XRD images of plated PSA fibers: SE images of (**a**) PSA@Ni, (**a′**) PSA@Ni@Ag, (**b**) PSA@Ni-P-B, (**b′**) PSA@Ni-P-B@Ag; and XRD patterns after (**c**) electroless plating, and (**c′**) the subsequent electroplating.

**Figure 3 nanomaterials-14-00115-f003:**
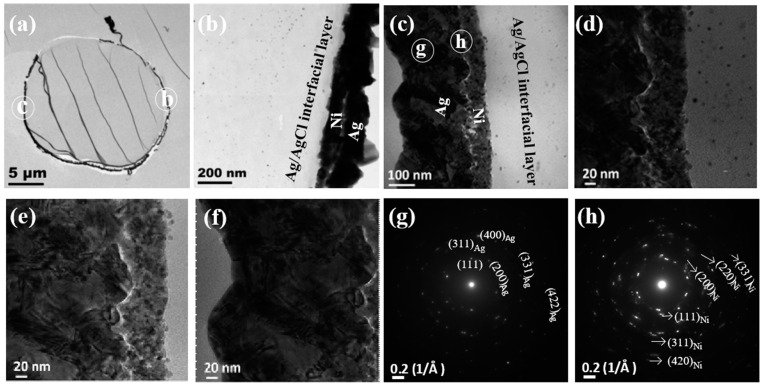
Cross-sectional TEM imaging of PSA@Ni@Ag: (**a**–**f**) bright field images and (**g**,**h**) SAED images.

**Figure 4 nanomaterials-14-00115-f004:**
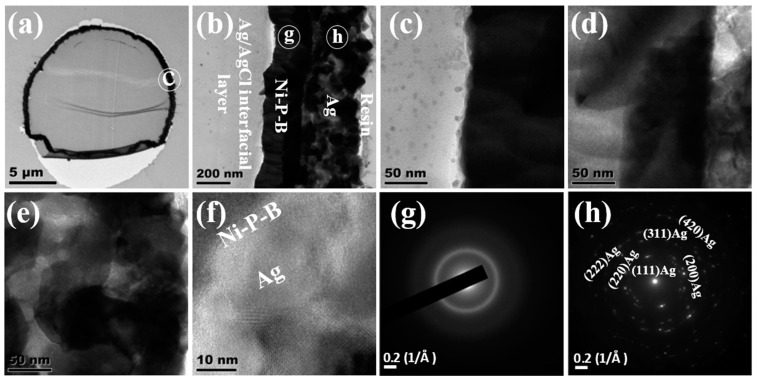
Cross-sectional TEM imaging of PSA@Ni-P-B@Ag: (**a**–**e**) bright field images, (**f**) high-resolution interface and lattice image, and (**g**,**h**) SAED images.

**Figure 5 nanomaterials-14-00115-f005:**
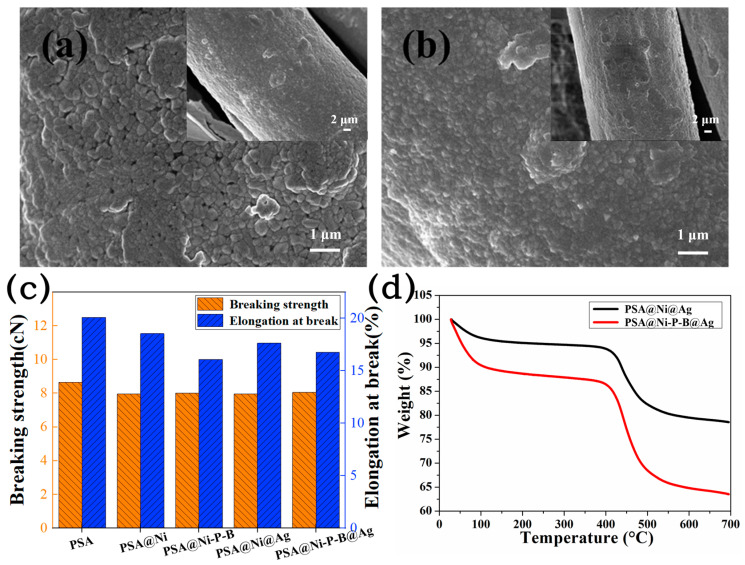
SEM images of (**a**) the PSA@Ni@Ag fiber and (**b**) the PSA@Ni-P-B@Ag fiber after ultrasonication in deionized water for 300 min; (**c**) tensile strength of the pristine and metallized fibers; (**d**) TGA curves of the PSA@Ni@Ag and the PSA@Ni-P-B@Ag fibers under a N_2_ atmosphere.

**Figure 6 nanomaterials-14-00115-f006:**
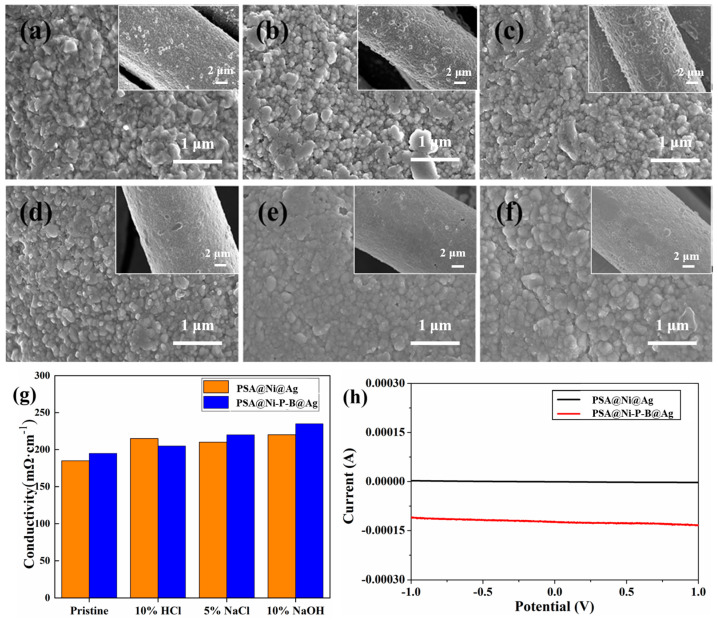
SEM images of (**a**–**c**) the PSA@Ni@Ag fiber and (**d**–**f**) the PSA@Ni-P-B@Ag fiber after immersion in (**a**,**d**) 5 wt.% NaCl solution, (**b**,**e**) 10 wt.% HCl solution, and (**c**,**f**) 10 wt.% NaOH solution for 24 h at 25 °C, respectively; (**g**) surface electric resistances of PSA@Ni@Ag and PSA@Ni-P-B@Ag fibers after chemical corrosion tests; (**h**) anodic polarization of the PSA@Ni@Ag and PSA@Ni-P-B@Ag fibers in 5 wt.% NaCl solution.

**Figure 7 nanomaterials-14-00115-f007:**
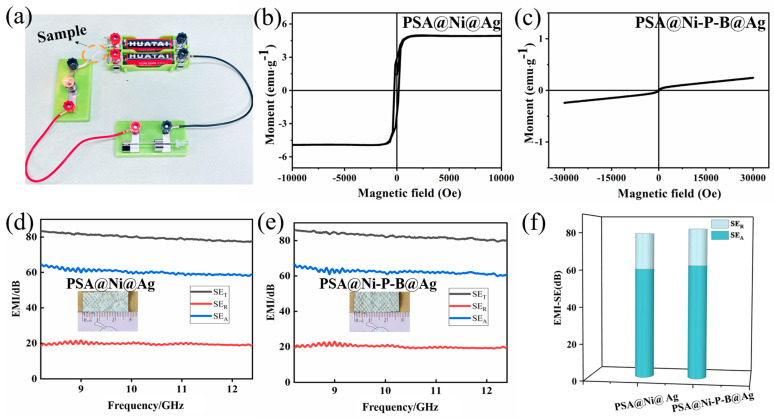
(**a**) Photograph of the bulb lit through the conductive fibers; the M−H curves of (**b**) the PSA@Ni@Ag fiber and (**c**) the PSA@Ni-P-B@Ag fiber; EMI shielding efficiency curves of (**d**) the PSA@Ni@Ag fabric, and (**e**) the PSA@Ni-P-B@Ag fabric in the X band (8–12 GHz); (**f**) average SE_A_ and SE_R_ of PSA@Ni@Ag and PSA@Ni-P-B@Ag in the X band.

**Figure 8 nanomaterials-14-00115-f008:**
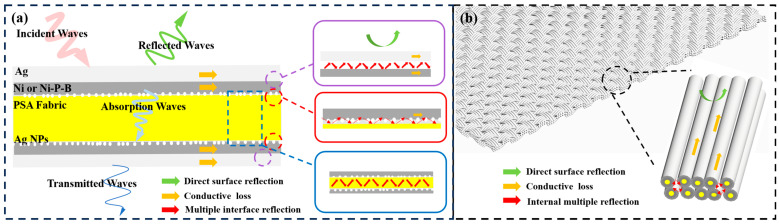
Schematic illustration of EMI shielding mechanisms for (**a**) a single metallized PSA fiber, and (**b**) the whole woven fabric.

**Table 1 nanomaterials-14-00115-t001:** Bath composition and conditions of both Ni and Ni-P-B plating.

Composition and Conditions	Ni Plating Solution	Ni-P-B Plating Solution
Ni(CH_3_COO)_2_·4H_2_O	15 g∙L^−1^	15 g∙L^−1^
Na_3_C_6_H_5_O_7_·2H_2_O	5 g∙L^−1^	20 g∙L^−1^
NaH_2_PO_2_·H_2_O	5 g∙L^−1^	15 g∙L^−1^
DMAB	0.2 g∙L^−1^	2 g∙L^−1^
SDS	10 mg∙L^−1^	10 mg∙L^−1^
pH (adjusted by ammonia)	9.5	8.5
t	60 min	60 min
T	60 °C	60 °C

**Table 2 nanomaterials-14-00115-t002:** Bath composition and conditions of Ag electroplating.

Composition	Concentration and Condition
AgNO_3_	20~30 g∙L^−1^
DMH	100~110 g∙L^−1^
K_2_CO_3_	50~60 g∙L^−1^
Stabilizer E	10 mL∙L^−1^
pH	11.00
T	60 min
T	30 °C

## Data Availability

The datasets used and/or analyzed in the current study are available from the corresponding author upon reasonable request.

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
