# Peer review of "Highly Conducting Surface-Silverized Aromatic Polysulfonamide (PSA) Fibers with Excellent Performance Prepared by Nano-Electroplating"

_nanomaterials, 2024, doi:10.3390/nano14010115_

Round 1
Reviewer 1 Report
Comments and Suggestions for Authors
A manuscript contains original research work in which the authors have developed efficient nanocoating of polysulfonamide (PSA) fibers characterized by high conductivity and high interference shielding effect. Surface coating of PSA fibers were obtained by electrolysis of Ni or Ni-P-B alloy followed by silver electroplating. The work presents reliable results and their reasonable interpretation. I recommend that this paper be accepted as presented, subject to one technical remark: in caption to Figure 8, the "e" position is missing, and the "c" position is used twice.
Comments on the Quality of English Language
Minor editing of English language required.
Reviewer 2 Report
Comments and Suggestions for Authors
The manuscript entitled, ‘Highly Conducting Surface-Silverized Aromatic Polysulfonamide (PSA) Fibers with Excellent Performance Prepared by Nano-Electroplating’ reported electroconducting fiber by nanoelectroplating. The article should be modified according to the following points;
1. The novelty of the article should be more emphasized.
2. Author wrote ‘Silver is widely used to construct EMI shielding…’; some examples should be provided on this ground.
3. There is a significant mass loss shown in TGA data. The cause is not clear. Better elaboration needed.
4. Did the author check the multiple reflections for the fibers? How that could be argued?
5. Some articles have significance and could be discussed with the help of following references:
(a) Ganguly, S., Das, P., Saha, A., Noked, M., Gedanken, A., & Margel, S. (2022). Mussel-inspired polynorepinephrine/MXene-based magnetic nanohybrid for electromagnetic interference shielding in X-band and strain-sensing performance. Langmuir, 38(12), 3936-3950.
(b) Wang, X. X., Zheng, Q., Zheng, Y. J., & Cao, M. S. (2023). Green EMI shielding: Dielectric/magnetic “genes” and design philosophy. Carbon, 206, 124-141.
Reviewer 3 Report
Comments and Suggestions for Authors
Overall this work is well written and designed. Listed below are a few comments and suggestions to help improve the presentation.
Page 2 line 71: You write "Electroless", do you mean "Electrodeless"? Please review throughout paper and correct if necessary.
Page 2 Line 95: You list a Sodium hypochlorite solution as having a pH of 3.0, Please review this as this solution should have a pH that is much higher than that listed.
Figure 2 (c) and (c'): The fonts for the labels are too small, please enlarge them so they are easier to read and readable when printed. This goes for all figures, please review and modify as necessary.
Figure 3 (b) and (c): The text used over the photos is unreadable. Please adjust the contrast.
Reviewer 4 Report
Comments and Suggestions for Authors
SUMMARY
In the paper, the authors presented a method for fabricating PSA@Ni@Ag and PSA@Ni-P-B@Ag composite fibers designed for robust electrical conduction and efficient electromagnetic interference (EMI) shielding. The process involved electroless plating to introduce inner Ni or Ni-P-B base layers onto the aromatic polysulfonamide (PSA) fibers, followed by electroplating to deposit outer Ag layers. The article is well-written and systematically organized, demonstrating a clear and concise presentation of the research conducted by the authors.
The experimental and analytical methods are thoroughly described, providing a comprehensive understanding of the processes involved in the fabrication and analysis of the composite fibers. The inclusion of detailed information about materials, plating conditions, and characterization techniques adds credibility to the study.
The results section is presented in a logical sequence, covering the morphology and structure of the composite fibers, their environmental stability, mechanical properties, and electromagnetic shielding performance. The authors use figures and images effectively to illustrate their findings, enhancing the clarity of the presented data.
The conclusions drawn from the research are well-supported by the results, and the authors discuss the implications of their findings for potential applications. The positive aspects of the proposed method, such as its simplicity, effectiveness, and the advantageous properties of the resulting fibers, are emphasized.
POSITIVE ASPECTS
1. The literature review sets the stage for the authors' research, providing context on the challenges in EMI shielding, the limitations of existing materials, and the potential of their proposed electroplating approach with 5,5-Dimethylhydantoin (DMH) as a complexing agent.
2. The authors describe the experimental procedures and analysis methods they employed. The combination of these methods and techniques allowed the authors to comprehensively investigate the morphology, structure, environmental stability, mechanical properties, and electromagnetic interference shielding performance of the proposed fibers.
3. The presented results suggest that these metalized PSA fibers have promising characteristics for EMI shielding applications, combining lightweight, high flexibility, good environmental stability, and high mechanical properties.
CONCERNS
The presented work is useful but has some concerns that need to be removed. My comments are merely editorial (of minor type). Points that must be addressed by authors are listed below:
Minor concerns
1. According to ISO 80000-1: 2009 standard, the symbol ºC for the degree Celsius shall be preceded by a space when expressing a Celsius temperature (page 2, line 56).
2. Correct the abbreviation used for reflection shielding effectiveness (page 10, line 307).
3. In the Materials and Methods section, add the type and manufacturer of the VNA used to measure scattering parameters of fabrics.
CONCLUSION
I find this article helpful. Regretfully, the paper cannot be accepted in its present form. The authors of the present article have to correct the issues.
